# Clinical responses to adoptive T-cell transfer can be modeled in an autologous immune-humanized mouse model

Henrik Jespersen [1], Mattias F. Lindberg[1], Marco Donia[2], Elin M.V. Söderberg[1], Rikke Andersen[2], Ulrich Keller[3,4], Lars Ny[1], Inge Marie Svane[2], Lisa M. Nilsson[1] & Jonas A. Nilsson[1]

Immune checkpoint inhibitors and adoptive cell transfer (ACT) of autologous tumor-infiltrating T cells have shown durable responses in patients with melanoma. To study ACT and immunotherapies in a humanized model, we have developed PDXv2.0 — a melanoma PDX model where tumor cells and tumor-infiltrating T cells from the same patient are transplanted sequentially in non-obese diabetic/severe combined immune-deficient/common gamma chain (NOG/NSG) knockout mouse. Key to T-cell survival/effect in this model is the continuous presence of interleukin-2 (IL-2). Tumors that grow in PDXv2.0 are eradicated if the autologous tumor cells and T cells come from a patient that exhibited an objective response to ACT in the clinic. However, T cells from patients that are non-responders to ACT cannot kill tumor cells in PDXv2.0. Taken together, PDXv2.0 provides the potential framework to further model genetically diverse human cancers for assessing the efficacy of immunotherapies as well as combination therapies.

[1] From the Sahlgrenska Translational Melanoma Group, Institute of Clinical Science, Departments of Surgery and Oncology, Sahlgrenska Cancer Center at University of Gothenburg and Sahlgrenska University Hospital, Gothenburg 405 30, Sweden. [2] Center for Cancer Immune Therapy, Department of Hematology and Department of Oncology, Herlev Hospital, University of Copenhagen, Herlev DK-2730, Denmark. [3] Internal Medicine III, Technische Universität München, Munich 81675, Germany. [4] German Cancer Consortium/Deutsches Krebsforschungszentrum, Heidelberg 69120, Germany. Henrik Jespersen and Mattias F. Lindberg contributed equally to this work. Correspondence and requests for materials should be addressed to J.A.N. (email: jonas.a.nilsson@surgery.gu.se)

The prognosis of metastatic cutaneous malignant melanoma has been historically dismal[1] but recent developments in targeted therapies and immune therapies have resulted in prolonged overall survival. The disease most often develops from sun-exposed areas of the skin and hence has a high mutational load with a strong ultraviolet (UV) signature[2, 3]. Besides the obvious risk of causing oncogenic mutations, UV radiation may also result in a tumor promoting inflammation[4]. On the other hand, a high mutation load also poses a risk for the incipient tumor cell to elicit an immune response by giving rise to expression of so-called neoantigens[5, 6]. Indeed, spontaneous

regressions of melanoma[7] have been described and it is possible that the 5–10% of patients with metastatic disease that were cured from their disease prior to checkpoint inhibitors and targeted therapies[1] exhibited a better immune profile. Owing to the well-established immunogenicity, melanoma has therefore been an attractive disease for studying immune evasion and immunotherapy. Several strategies to boost immunity against melanoma have been tested including vaccines[8, 9], interleukin-2 (IL-2)[10], interferons[11] and histamine but with limited success[12]. Instead, cell-based techniques such as adoptive T-cell transfer (ACT)[13] or immune checkpoint inhibitory antibodies have

**Fig. 1** Effective *in vitro* cytotoxicity does not result in effective anti-tumoral activity in NOG mice. **a** Schematic representation of the humanization process (PDXv2.0). **b** Melanoma cells from patient #33 (MM33) were transduced with a luciferase lentivirus. Cells were plated in a 96-well plate and were mixed with post-REP TILs from the original tumor. Luciferin was added to the media and the viability of the luciferase-expressing tumor cells was measured in a luminometer. The experiment was done in triplicates and the error bar represents ± SEM. **c** MM33 cells were transplanted into ten NOG mice. 1 week after transplantation, mice were randomized into two groups, one of which received a tail vein injection with autologous TILs. 45,000 U IL-2 was given daily for 3 days following TIL injection and thereafter twice weekly during 3 weeks. Tumor growth was measured using calipers. *P*-values are from a multiple *t*-test analysis. **d**, **e** When robust progression was noted in all mice, the mice with the slowest and fastest growing tumors in each group were sacrificed and the tumors were analyzed by immunohistochemistry (**d**, bar = 50 µm) or flow cytometry **e** for indicated markers. When PD1 expression was observed, anti-PD1 antibody pembroluzimab was given to the mice twice weekly. The experiments shown were performed once

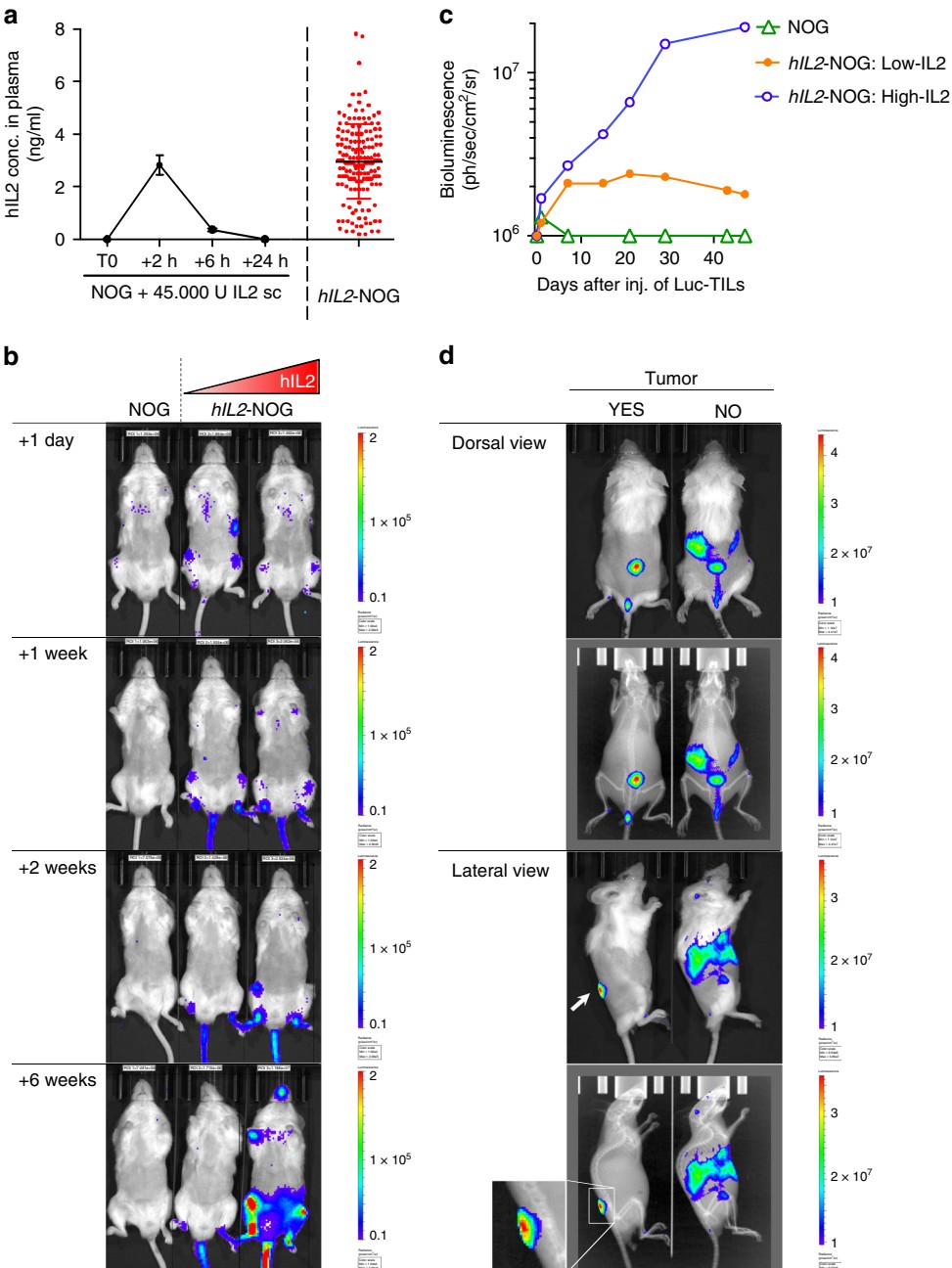

**Fig. 2** Transgenic expression of human IL-2 enables NOG mice to support viability of TILs. **a** Changes in IL-2 plasma levels in NOG mice injected with 45,000 U IL-2 over time and steady-state levels of IL-2 in *hIL2*-NOG mice. **b** In vivo bioluminescence images over time of two *hIL2*-NOG transgenic mice with different serum levels of hIL-2 injected with MM33 TILs labeled with a luciferase lentivirus. **c** Quantification of radiance of luciferase-expressing MM33 TILs growing in mice (from **b**) with different serum levels of IL-2. **d** In vivo bioluminescence images from different angles of tumor-bearing or non-tumor-bearing *hIL2*-NOG transgenic mice injected with luciferase-expressing MM33 TILs. The experiments shown here were performed once

yielded the most promising results[14–16]. Both antibodies directed against cytotoxic T-lymphocyte-associated protein 4 (CTLA4; ipilimumab) or programmed cell death protein-1 (PD1; nivolumab and pembrolizumab) are approved for use in melanoma patients. Although the response rates are significantly lower as compared to BRAF- and MEK-targeted therapies[17–20], the responses are often more durable[21]. An optimal treatment should therefore provide the response rates of targeted therapies and the durability of immunotherapies. If a monotherapy cannot achieve this, perhaps a combination therapy can. Indeed, combination therapy between BRAF inhibitors and immune therapy (anti-PD1) has been shown to be superior compared to monotherapies,

at least in genetically engineered mouse models (GEMMs)[22, 23]. However, if concomitant combination is tolerable in the clinic, or if not, in which schedule the combination should be used is not completely understood. An additional outstanding question is also whether combinations of different types of immune therapies such as checkpoint inhibitors and ACT would benefit patients.

The NOD-SCID-IL2-receptor common gamma chain knock-out (NOG or NSG) mouse has revolutionized the ability to grow tumor-grafts from patients[24]. The lack of all lymphocytes, including NK cells, and a polymorphism in SIRPa, the CD47 receptor, makes the model particularly amendable for growing human cells[25]. Melanoma grafts grow especially well as patient-

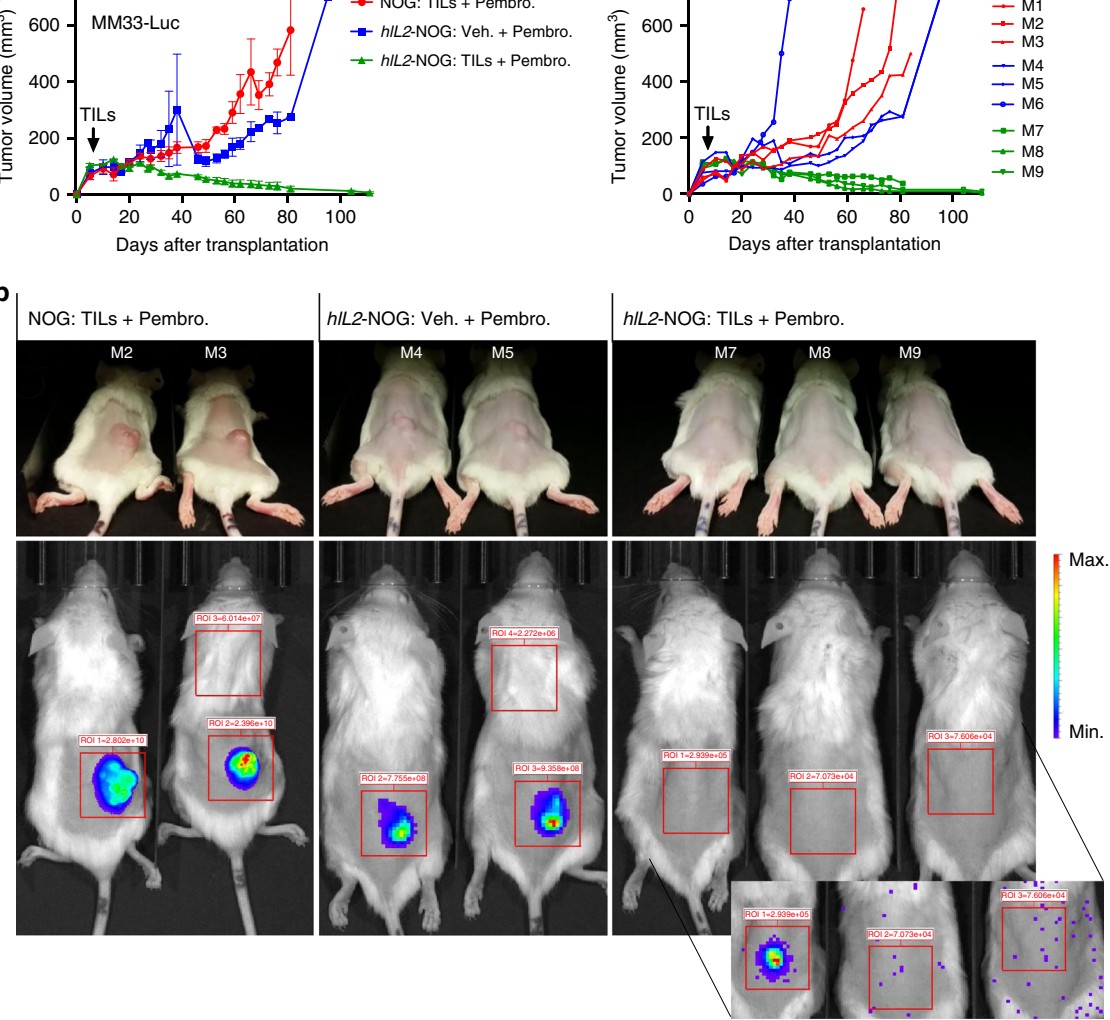

**Fig. 3** Transgenic expression of human IL2 enables TIL-mediated tumor eradication in NOG mice. **a** Six *hIL2*-NOG transgenic mice and three NOG mice were transplanted with luciferase labeled MM33 tumor cells. After tumor growth had been confirmed by IVIS imaging, all NOG mice, and three of the *hIL2*-NOG mice, were injected with autologous REP TILs and tumor growth was measured with calipers. Shown to the right are individual mice. **b** IVIS imaging to investigate if mice are disease free. *Inset*, very prolonged exposure yielding mostly spark signals from static electricity. The experiment described here was performed once, follow-up experiments confirming the data are in Fig. 4 and Supplementary Fig. 5

derived xenografts (PDXs) with near complete take rate[24, 26]. Yet, current melanoma PDXs are not suitable for studies of immune therapy studies. The aim of this study was to investigate if the effect of ACT in patients could be modeled in humanized mice in a predictive manner and if therapeutic effects would be enhanced by anti-PD1 therapy. Here we generate a novel model, PDXv2.0, by extended humanization of the NOG mouse with tumor-infiltrating T lymphocytes and human *IL2*.

### Results

**T cells can accumulate in autologous tumors in NOG mice.** Most melanoma tumors grow while being in the presence of tumor-infiltrating T-lymphocytes (TILs). Under normal conditions, these TILs do not mediate tumor regression, e.g., because of upregulation of PD-L1-mediated immune suppression (exhaustion)[27, 28]. In vivo, anti-PD-1 antibodies reactivate TILs causing tumor regression[15, 16]. Interestingly, TILs can also be re-activated in vitro by culturing pieces of melanoma biopsies in media containing the T-cell growth factor IL-2[29, 30]. The re-activated TILs can be expanded to large quantities, and can mediate

durable complete tumor regressions, when re-infused back into the individual patients[13]. We hypothesized that in vitro expanded TILs should be able to kill autologous tumors in NOG mice when adoptively transferred. If so, we would also be able to create a model, PDXv2.0 (Fig. 1a), which could be used to study immune therapy and combination therapies.

We used a rapid expansion protocol (REP) to expand TILs from patient #33. These TILs came from a metastatic melanoma (MM33) and were characterized by their very high in vitro cytolytic capacity as measured by a viability assay and release of interferon gamma (IFNγ) in co-cultures of TILs and tumor cells (Fig. 1b). Immune phenotyping suggested that the majority of the REP-TILs were cytotoxic effector memory cells (Supplementary Fig. 1). We injected MM33 tumor cells into 10 NOG mice and monitored tumor growth by caliper measurement. When the cells had formed palpable tumors, we injected $20\times10^6$ autologous MM33 TILs per mouse into five mice via the tail vein and treated the mice with recombinant human IL-2, akin to the protocol used in patients undergoing adoptive T-cell therapy. However, injection of TILs barely caused any effect on tumor growth (Fig. 1c).

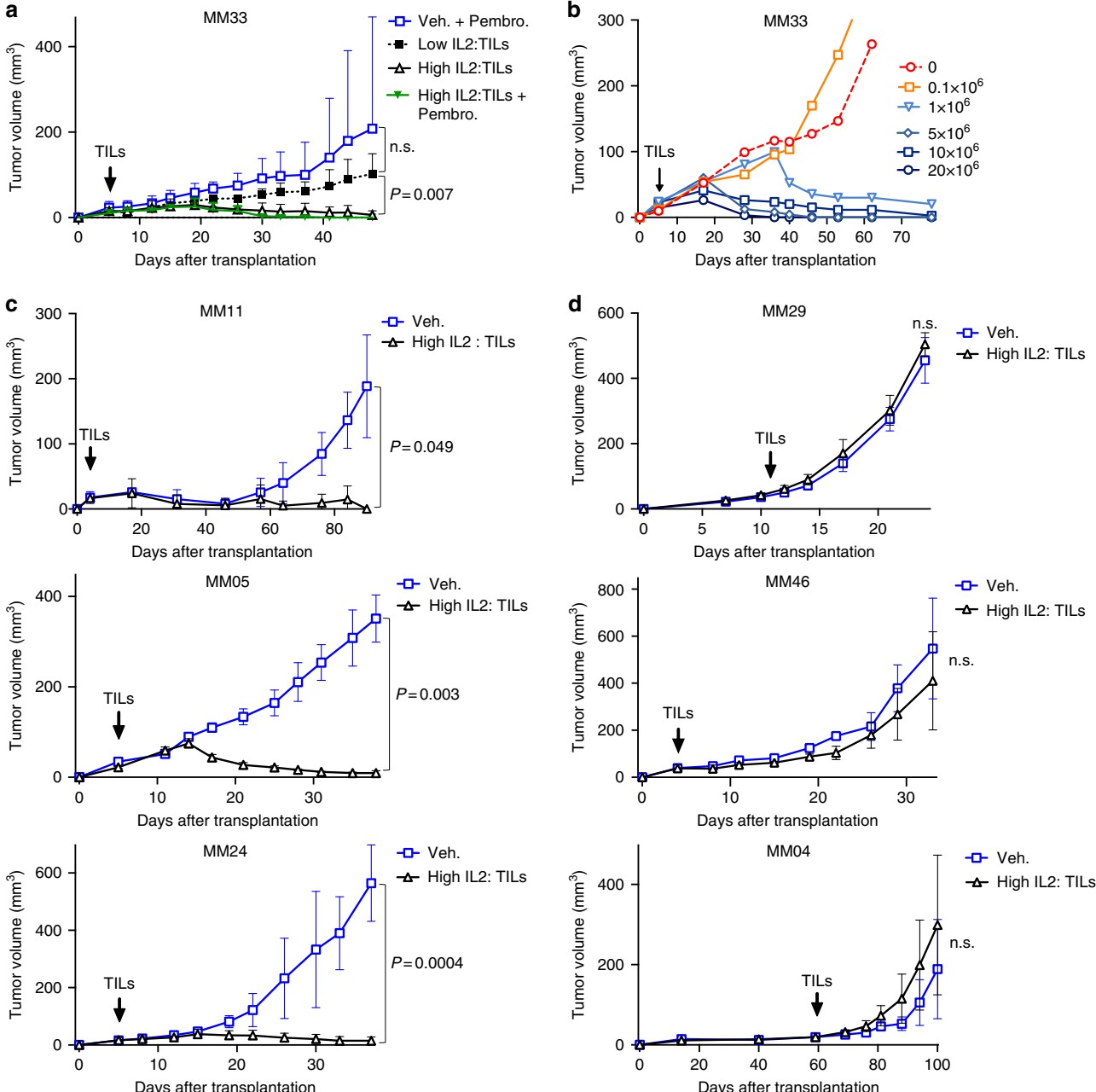

**Fig. 4** ACT in mice with high levels of plasma hIL-2 correlates with responses in patients. **a** Tumor growth curves of MM33 in *hIL2*-NOG mice expressing high (>2 ng/ml) or low levels of hIL-2 (<1.5 ng/ml), treated either with PBS or TILs in the presence or absence of pembrolizumab. **b** Dose titration response of MM33 to different amounts of injected TILs (from 0.1×10⁶ to 20×10⁶ TILs) in *hIL2*-NOG mice with high levels of hIL2. **c**, **d** Tumor growth curves of cells from three responders in the clinic (**c**: MM11, MM24 and MM05) and from two non-responders (**d**: MM04, MM46 and MM29) in *hIL2*-NOG mice with high levels of hIL-2 treated either with PBS or TILs. Data are caliper measurements (MM33/MM24/MM05/MM04, *n* = 4 per group; MM11/MM29/MM46, *n* = 3 per group). *P*-values are from multiple *t*-test (with Sidak corrections) for tumor growth curves. Experiment in a was performed twice, a replicate shown in Supplementary Fig. 5. Experiments in **b**, **c** were performed once except for MM24 where a replicate experiment is shown in Supplementary Fig. 5

To investigate if TILs injected into NOG mice track the tumor cells we sacrificed two mice per group with the smallest and the biggest tumors, and analyzed their tumors by immunohisto-chemistry (IHC) and flow cytometry. The TILs found in the tumor were T cells as evident from the presence of CD3+ cells by IHC (Fig. 1d). However, even though the original in vitro expanded and subsequently transplanted TILs did not express the negative immune checkpoint protein PD-1, in vivo growth resulted in induction of surface PD-1 expression of the TILs (Fig. 1e), suggesting that the tumor cells communicated with the

TILs. We therefore treated the remaining mice in the tumor growth study with clinical grade anti-PD-1 antibody pembroli-zumab but this did not result in the desired tumor regression (Fig. 1c).

**TILs do not cause tumor regression in NOG mice.** Despite MM33 TILs were able to kill MM33 tumor cells in vitro, that they were trafficking to the tumor site in mice, and that anti-PD-1 antibodies were administered, no regression of tumors was

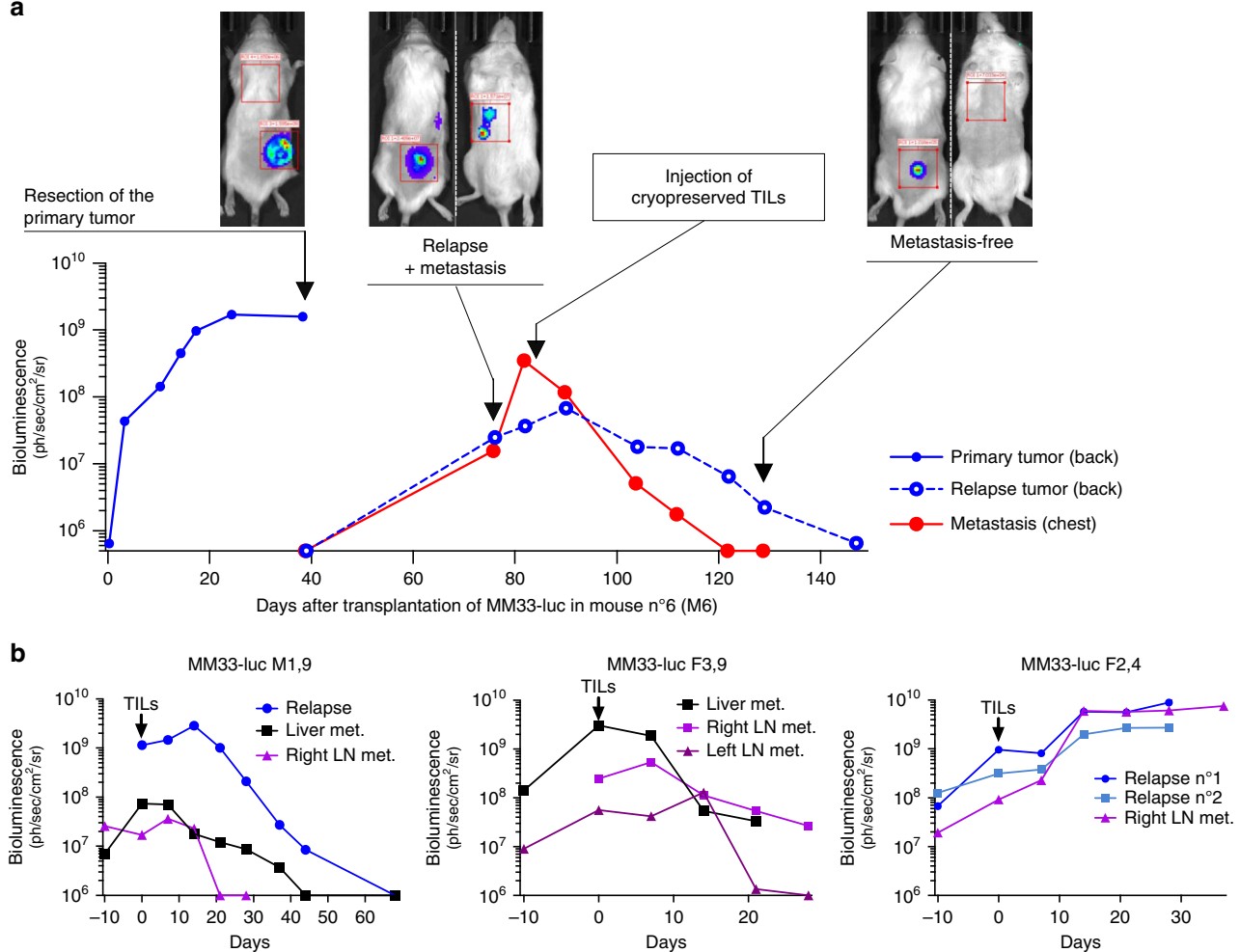

**Fig. 5** Transgenic expression of human IL2 enables TIL-mediated tumor eradication of metastases in NOG mice. Transgenic *hIL2*-NOG mice transplanted with luciferase labeled MM33 tumor cells were allowed to develop subcutaneous tumors until tumors reached >500 mm$^3$. The tumors were surgically removed and relapse was followed using the IVIS imager. When metastases had appeared TILs were injected and responses were measured by IVIS imaging. **a** A representative mouse showing the disease progression. **b** IVIS imaging of mice with additional sites of metastasis of melanoma. Graphs are labeled with gender of mouse (F or M) and corresponding plasma level of hIL-2 (ng/ml)

observed in this model. One possibility for ineffectiveness of the described procedure was that we had used TILs that would have been incapable of causing effects in adoptive T-cell therapy in patients as well. To test that hypothesis, we developed tumor cells and corresponding TILs from patients that had participated in a clinical trial of adoptive T-cell therapy[31], with different levels of response to TIL therapy (Supplementary Fig. 2a). We first established xenografts from the tumor cells and noted that most grew at a rate feasible for testing in vivo (Supplementary Fig. 2b). We therefore repeated our procedure (Fig. 1a) with the exception that we randomized 15 mice into three different groups; one group transplanted with tumor cells and that received pembrolizumab and IL-2; a second group transplanted with tumor cells and TILs and that received IL-2, and a third group transplanted with tumor cells and TILs and that received pembrolizumab and IL-2. TILs from the completely responding patient (#11) and non-responding patient (#29) failed to affect the growth of the autologous tumor cells in mice. A slightly protracted disease progression—but limited impact on overall survival—was observed in tumor-bearing mice transplanted with MM24 TILs (Supplementary Fig. 3). In the clinic, patient #24 exhibited a partial response following infusion with TILs but underwent additional surgery to remove the remainder of the

tumor. After this, the patient has remained tumor free for over 3 years (Supplementary Fig. 2a). Finally, there was a weak tumor suppressive effect of MM46 TILs but only when mice were also treated with pembrolizumab (Supplementary Fig. 3). Taken together, TILs can delay tumor growth in some mice but cannot achieve regressions in NOG mice—even if the TILs came from patients that responded to adoptive cell transfer (ACT) in the clinical trial.

**IL-2 is essential for tumor eradication in NOG mice.** The fact that we were unable to achieve appreciable tumor regressions by adoptive TIL transfer into NOG mice made us hypothesize that a crucial factor was missing or non-functional in mice, or that the tumor was utilizing more ways to evade immunity besides PD-L1 expression. IL-2 is essential for T-cell expansion and it is possible that the clinical grade IL-2 used here had a too short half-life or the dosing schedule or dose selected[32] was too low. Indeed, as published before[33], we observed that subcutaneous injection of human IL-2 resulted in peak levels around 2 hrs, which were vanished by 8 h (Fig. 2a). To circumvent this we obtained *hIL2*-NOG transgenic mice that overexpress IL-2 by virtue of a CMV-driven human *IL2* transgene[34]. Measuring plasma levels of IL-2

by enzyme-linked immunosorbent assay (ELISA) in offspring of these transgenic mice revealed a large spread from basically no IL-2 production to levels above 5 ng/ml, comparable to the peak levels achieved by human IL-2 injection (Fig. 2a). To investigate if these mice could support human T cells we transduced MM33 TILs with a luciferase-GFP virus and injected them into NOG or hIL2-NOG transgenic mice. Reassuringly, TILs survived longer in hIL2-NOG than in NOG mice, even in low-expressing transgenics, causing disseminated growth (Fig. 2b, c). Transplant of TILs into tumor-bearing mice made the TILs accumulate at the site of the tumor, suggesting a homing mechanism or that the tumor can prevent systemic spread of T lymphocytes (Fig. 2d). The TILs could also be detected by flow cytometry in tumor homogenates and the majority of T cells were CD8+ effector memory cells, whereas in peripheral blood the CD4+ cells predominated (Supplementary Fig. 4).

To investigate if IL-2 was a limiting factor for curative effects of ACT, we labeled MM33 tumor cells with luciferase and transplanted them into hIL2-NOG mice. In contrast to the effects in NOG mice (Fig. 1c), MM33 TILs injected in hIL2-NOG mice were able to cause regression of MM33 tumor cells in mice treated with pembrolizumab (Fig. 3a). Treatment was continued until no bioluminescence signal could be detected (Fig. 3b), suggesting the mice had undergone a complete response (CR). In a repeat experiment we observed one hIL2-NOG mice carrying MM33 tumor mouse that did not achieve a CR by MM TILs. This mouse turned out to have lower serum IL-2 levels (Supplementary Fig. 5a). We therefore repeated the experiment using mice with >2 ng/ml IL-2 or <1.5 ng/ml IL-2. This experiment shows that 2 ng/ml or above is optimal to achieve CR in MM33 ACT experiments (Fig. 4a). As expected, we also noted that addition of pembrolizumab did not alter the already robust response to ACT in hIL2-NOG mice (Supplementary Figs. 4a and 5a). In a dose–response experiment we determined the minimal amount of TILs to be used in successful ACT was $10^6$ cells (Fig. 4b).

Next, we repeated the experiment using tumors and TILs from complete responding patient #24. Two out of three mice transplanted with MM24 tumor cells were as responsive to MM24 TILs as those transplanted with MM33 tumor cells and TILs but in the third mouse, regressions was not observed (Supplementary Fig. 5b). A follow-up repeat experiment, confirmed that also the MM24 model required a minimum concentration of 1.5 ng/ml plasma IL-2 to achieve regressions (Fig. 4c). The IL-2 dependence prompted us to re-address the original experiment (Fig. 1c) in NOG mice that did not result in tumor regression. Indeed, establishing a higher dose plasma level (Supplementary Fig. 5c) and daily injections of recombinant human IL-2 for 16 days resulted in tumor regressions (Supplementary Fig. 5d). Contrary to the hIL2-NOG mouse model though, the response was not durable when injections were stopped and only one mouse out of three was cured.

Confident that we had modeled ACT in mice, we were interested in assessing whether or not the PDXv2.0 model correlated with responses in the clinic. To that end we used six different patients TILs and tumors (Supplementary Fig. 2a, b), three responders' cells and three non-responders' cells. Unlike that seen in NOG mice, the hIL2-NOG mice transplanted with MM24, MM11 and MM04 tumor cells all responded to therapy with autologous TILs, whereas those transplanted with MM29, MM46 and MM05 tumor cells did not (Fig. 4c, d).

Melanoma can metastasize to many distant sites in patients. To investigate if ACT in mice could achieve regression of spontaneous metastasis MM33 tumors were surgically resected from in hIL2-NOG mice that did not receive MM33 TILs. As expected, surgical removal of the tumors caused a drop in bioluminescence but relapse emerged around the area of resection

as well as in distant sites (Fig. 5a, b). When MM33 TILs were injected, it resulted in clearance of metastatic disease from most sites in three out of four mice. Taken together, ACT in hIL2-NOG mice models ACT in patients not only when the tumors grow subcutaneously.

## Discussion

Here we model ACT in mice and describe a new PDX model termed PDXv2.0 that is useful to study immune therapy in humanized mice. To our knowledge this model is the first to show CRs of human tumor cells by autologous immune cells in a immune compromised mouse model—thus recapitulating the robust effects observed in some patients with malignant melanoma treated with immune therapy such as ACT or anti-PD1 antibodies. Besides developing the model, we also unravel biological insights into T-cell immune therapy. First, ACT in mice correlates with results of the same patients' cells in the clinic. Second, transplanting autologous T cells to tumor-bearing NOG mice is not sufficient to achieve tumor eradication unless IL-2 is supplied continuously, which is facilitated by using the hIL2-NOG mice. Third, anti-PD1 therapy does not alleviate the need of IL-2 in NOG mice. Our study therefore offers a proof-of-concept that a PDX model could be used to screen patients' tumors and TILs to predict responses to ACT and perhaps other types of immune therapy, although to firmly establish this a larger and prospective study is needed. A similar concept has already gained traction for predicting targeted therapy[35–37] but since PDX models have note been useful in immune therapy studies to date, this has not been tested hitherto.

A major finding of this paper is the apparent dependency of TILs on continuous supply of IL-2. This is notable considering the TIL therapy trials in patients that are currently performed to assess the appropriate dosing of IL-2. The side effects observed with IL-2 in patients historically are not trivial, often requiring hospitalization. At least two currently registered trials are testing the hypothesis that high-dose bolus dose IL2 can be avoided. In one trial (ClinicalTrials.gov Identifier: NCT01995344), a lower dose is tested whereas in another (ClinicalTrials.gov Identifier: NCT01468818), which was recently terminated due to low accrual rates, was testing to omit IL-2 altogether. Moreover, a small pilot study (NCT00937625) has shown that CRs in patients can be obtained by TIL therapy using daily low-dose IL-2 injections[38]. In the recently published phase II trial (NCT00937625) of patients receiving TIL therapy and a lower, but continuous dose of IL-2, similar response rates as trials using high-dose IL-2, was demonstrated[31]. Learning from the mouse model described here it seems plausible that omitting IL-2 would be unadvisable. Instead, means to limit IL-2 toxicity by lower but continuous dosing, or by using alternative forms of hIL-2 such as the so-called IL-2 superkine could make TIL therapy more accessible to the general oncology clinic[39]. The method described here can be used to study potential improvements and guide the technical and clinical development of immune therapy against melanoma and possibly other diagnoses.

The PDXv2.0 model may also be used to study resistance mechanisms to immune therapy and therapeutic responses to combination therapies consisting of cancer immune therapy and targeted therapies or chemotherapy. A clear advantage of the PDXv2.0 model over, e.g., GEMMs is that the tumor cells and TILs used can come from a biobank of many different patients or lesions from the same patient, thus recapitulating the heterogeneous nature of the disease. Furthermore, the tumor cells being human also enable the direct test of clinical agents that may have been developed against human proteins. The drawbacks are however potentially equally many. For instance, at this time we

are unaware to what extent the immune suppressive environment in the tumor of the patient is represented in the model. The stroma and myeloid cells in our model is murine and hence is unlikely to exert immune suppressive power. Another drawback is that we were never able to show a benefit of anti-PD1 since the model was either curative or the tumors did not respond to ACT at all. It is therefore possible that altering the protocol and use fewer T cells is needed if combination studies of ACT and checkpoint inhibitors such as anti-PD1 are to be successful.

Humanizing mice with respect to both tumor and immune cells has been tested before using various approaches. In these models, either bone marrow aspirates, CD34+ hematopoietic stem cell injection (Hu-CD34-NSG)[40], or peripheral blood mononuclear cells (PBMC) has been transplanted into immune-deficient mouse models together with either autologous cells or traditional cell lines[41, 42]. Although these models are also feasible, the concern is primarily dealing with T-cell education and lack of CRs. Besides being less effective, erroneously trained T cells and other cells from the hematopoietic system frequently cause graft-vs-host disease (GVHD). Indeed, a very recent GVHD study demonstrates that NOG mice transplanted with human CD4+ T cells (but not CD8+ cells) from PBMC develop serious skin phenotypes, alopecia, weight loss and death[43]. They also demonstrate that CD8+ expand vigorously in hIL2-NOG mice and cause weight loss and death by 8 weeks after transplantation. Many of the experiments reported in our study were conducted for >10 weeks and besides observing weight loss or large spleens in some mice (Supplementary Fig. 6) that had been cured from melanoma we never observe the dramatic skin phenotypes of GVHD. On the other hand, as shown in Fig. 2, TILs home to the tumor so in the absence of a tumor or target, perhaps a lymphoproliferative disease can occur in hIL2-NOG mice. We hypothesize that PDXv2.0 does not develop GVHD for at least three reasons: (i) the tumor cells transplanted does not contain immune cells; (ii) the TILs have been selected in the patients to recognize the tumor cells and may therefore be specific; and (iii) the TIL cultures do not contain B-cells, macrophages or other immune cells that also constitute major culprits in GVHD[44, 45].

Future head-to-head comparisons are warranted to investigate under which circumstance one model should be chosen over the other. For instance, in our case we most often do not observe any major effect of anti-PD1 therapy since the effect of ACT is sufficient to cause tumor clearance in responders, and that anti-PD1 therapy is not able to convert a non-responder into a responder. Therefore, to study anti-PD1 therapy other models where the T cells are less efficient at tumor eradication should likely be used[46]. Notably, the current method developed demonstrates that it is possible to achieve CRs to TIL therapy in immune compromised mice thereby setting a new standard to aim for in model development.

## Methods

**Generation of TILs.** After informed consent and IRB approval (he Ethical Committee of the Capital region of Denmark), patient samples were obtained by surgical specimen collection from patients with metastatic melanoma as part of a clinical trial[31]. A metastatic lesion was cut into fragments of 1–2 mm$^2$ that were placed in separate wells in a 24 well-plate (Nunc) with 2 ml of culture medium (90% RPMI 1640 (Invitrogen), 10% heat inactivated Human AB serum (HS, Sigma-Aldrich), 6000 IU/ml recombinant human IL-2 (Aldesleukin, Novartis), penicillin, streptomycin and fungizone (Bristol-Myers Squibb). Young TIL (yTIL) cultures were obtained by pooling TILs from each fragment as previously described[47–49], before being cryopreserved. For use in PDX-ACT experiments, yTILs were expanded using a standard small-scale REP[49]. In short, irradiated (40 Gy) allogeneic feeder cells (2×10$^7$), CD3 antibody (clone: OKT3) (30 ng/ml) (Miltenyi), 10 ml culture medium, 10 ml REP medium (AIM-V, Invitrogen) supplemented with 10% HS and 6000 IU/ml IL-2) and yTILs (1×10$^5$) were mixed in a 25-cm$^2$ tissue culture flask. Flasks were incubated upright at 37 °C in 5% CO$_2$. On day 5, half of the medium was replaced. On day 7 and every day thereafter, cells were split into further flasks with additional REP-medium as needed to maintain cell

densities around 1–2×10$^6$ cells/ml. On day 10–14, cells were harvested and resuspended at 2×10$^8$ cells/ml of PBS (Lonza) supplemented with 300 IU/ml of IL-2, before being intravenously transplanted to mice (100 μl, 20×10$^6$ TILs per mouse). To enable in vivo tracking of the TILs, yTILs were transduced with lentiviral firefly luciferase immediately before initiating the REP.

**Cell lines.** All cell lines were established at the Center for Cancer Immune Therapy, Herlev, Denmark. Autologous melanoma cell lines were identified by their morphological appearance and growth pattern and established from the same tumor as the TILs by serial passage of adherent cells in RPMI 1640 supplemented with 10% fetal bovine serum (FBS) and 500 ng/ml of Solu-cortef, as previously described[49]. The cell lines were transduced with pHAGE-GFP-luciferase lentivirus and clones were obtained by single-cell cloning in 96-well plates. All cell lines were maintained in complete medium (RPMI-1640 supplemented with 10% FBS, glutamine and gentamicin), were regularly screened for mycoplasma by PCR and cultured at 37 °C with 5% CO$_2$.

**In vitro antitumor reactivity and cytotoxicity.** Luciferase cell lines (clones or pool of transduced cells) were plated at 5,000 cells/well in a 96 white well plate (Sarstedt) and TILs were immediately added onto them at different ratios. After 24 h of incubation at 37 °C and 5% CO$_2$, 20 μl of supernatant was collected for analysis of IFN-γ release by ELISA (as per manufacturers protocol, Diaclone). Luminescence was assessed with a Perkin Elmer Victor 3 plate reader, after adding 100 μl of luciferin (300 μg/ml) onto the cells.

**Mouse experiments.** All animal experiments were performed in accordance with EU directive 2010/63 (regional animal ethics committee of Gothenburg #2014-36). The first passage of the different PDXes (so called 'P1') was obtained by injecting 2×10$^5$ cells mixed with equal volume of Matrigel (Corning) subcutaneously at the flank of 6–15 week old immunocompromised, non-obese severe combined immune-deficient interleukin-2 chain receptor γ knockout mice (NOG mice; Taconic). P1 tumors with volumes above 200 mm$^3$ were then extracted, mechanically dissociated with and filtered through a cell strainer, washed with and resuspended in RPMI-1640 and finally transplanted in P2 treatment mice, either NOG or human IL-2 transgenic NOG (hIL2-NOG) mice (Taconic). When luciferase tumor cells were used, 2–4×10$^5$ cells mixed with Matrigel were transplanted.

Before treatment blood samples were collected from the hind leg vein (vena saphena), for analysis of plasma hIL-2 levels by ELISA as per manufacturers recommendation (BD Biosciences). Between 5 and 25 days following transplantation in P2 mice, mice were divided into treatment groups of equal proportions of tumor sizes and 20×10$^6$ autologous TILs were administered to the predefined mice through tail vein injection. Subcutaneous administrations of 2.75 μg IL-2 (45,000 IU, from Novartis) were given to the NOG mice at the same time as the TILs were injected, during the next two consecutive days, and then bi-weekly for 3 weeks. Alternatively, 2.75 μg IL-2 (≥45,000 IU, from Peprotech) was given once daily for 16 days. Pembrolizumab (MK-3475, Merck Sharp & Dohme) diluted in PBS was given intraperitoneally bi-weekly (250 μg in 100 μl).

Tumor volumes were measured by caliper at regular time points by an operator that was not blinded to the group allocation. For in vivo visualization of Luc-tumors or Luc-TIL distribution, mice were injected intraperitoneally with 100 μl luciferin (30 mg/ml, PerkinElmer) before being anesthetized with isoflurane. Luminescence quantifications were acquired in the regions of interest (tumor area or whole animal body in case of Luc-TILs) using the IVIS Lumina Series II (PerkinElmer) and associated software.

**Immunohistochemistry.** Xenograft tissues were fixed in 4% formalin, dehydrated and embedded in paraffin. Sections of 4 μm were mounted onto positively charged glass slides and dried overnight (ON) at 37 °C. The slides were rehydrated and antigen retrieval performed by pressure cooking in a citrate buffer, followed by quenching in 3% H$_2$O$_2$ in methanol for 10 min and blocking in 2,5% normal horse serum for 30 min. Slides were stained with CD3 (clone: A0452, Dako, dilution 1:500) and PD-L1 (clone: E1L3N, Cell Signaling Technology, dilution 1:200) antibodies ON at 4 °C, followed by 1 h incubation with secondary HRP-linked anti-rabbit antibody (MP-7401, Vector Laboratories). ABC peroxidase detection system (Vector Laboratories) was used following the manufacturer's instructions and Diaminobenzidine (DAB) chromogen developed. The slides were finally counter-stained with hematoxylin, dehydrated and mounted with Pertex.

**Flow cytometry.** TILs were washed once, resuspended at 200,000 cells/50 μl in MACS-buffer and stained with the following fluorochrome-conjugated antibodies: Alexa Fluor 488 anti-human CD3 (clone: UCHT1), APC anti-human CD279 (PD1) (clone: EH12.2H7), PE/Cy7 anti-human Tim-3 (clone F38-2E2) (BioLegend). Cells were then washed and analyzed on an Accuri C6 flow cytometer equipped with the BD Accuri C6 software.

For detailed immune phenotypning, cells were analyzed by flow cytometry at the department of Clinical Immunology and Transfusion Medicine, Sahlgrenska University Hospital (accredited to EN ISO 15189). For ex vivo characterization of TILs in ACT treated mice, TILs were injected in hIL2-NOG mice bearing established autologous tumors. Tumors were harvested 40 days after injection of

TILs, homogenized using a tissue chopper (McIlwain) and filtered repeatedly through 70 and 40 μm cell strainers before analysis. For PBMCs, blood was harvested from anesthetized mice through cardiac puncture 48 days after injection of TILs. PBMCs were then isolated using Ficoll-Paque before analysis.

**Statistical analysis**. Values are presented as mean ± S.E.M. when data is combined. For statistical analyses, we used Graphpad Prism software: multiple *t*-test (with Sidak corrections) for tumor growth curves and the log-rank test for survival. All mouse experiments contained 3–5 mice per group (exact number specified in figure legends). The sample size of five was pre-determined to be able to detect a >30% suppression of growth in the experiments with NOG mice (power 0.80, common standard deviation of 20%) but since we were able to achieve curative effects in the hIL2-NOG mice, sample size was decreased to three (power >0.9). $p < 0.05$ was considered statistically significant. Since all mouse strains were generated and maintained on the same original inbred background (NOG), the variation within each data set obtained by experiments with mice was assumed to be similar. However, experimental assessment of variance was performed and revealed a difference in effects of ACT based on levels of IL-2 (Fig. 4a, b and Supplementary Fig. 5a, b). No randomization was used in these studies. We were not blinded to group allocations during animal experiments.

**Data availability**. The authors declare that data supporting the findings of this study are available within the article and its Supplementary Information files and from the corresponding author upon reasonable request.

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

## Acknowledgements

We thank Sofia Nordstrand for assistance with animal experiments, Bengt Andersson and Ulla Johansson for immune phenotyping, Erik Larsson and Joakim Karlsson for statistical discussions, and Gordon Freeman and Arlene Sharpe for insightful comments on our study. This work was supported by the Swedish Cancer Society, the Swedish Research Council, the Region Västra Götaland (Sahlgrenska University Hospital, Gothenburg), the Knut and Alice Wallenberg Foundation, the Familjen Erling-Persson Foundation, the IngaBritt and Arne Lundberg Foundation and BioCARE—a National Strategic Cancer Research Program at University of Gothenburg (to J.A.N.), and from the Assar Gabrielsson Foundation, the W&M Lundgren Foundation and Sahlgrenska Universitetssjukhusets stiftelser (Sahlgrenska University Hospital, Gothenburg) (to E.M.V.S. and M.F.L.).

## Author contributions

H.J., M.F.L. and E.M.V.S. performed experiments. M.D. developed all cell lines and TILs. L.M.N. handled the mouse colonies. I.M.S. and U.K. contributed essential reagents. R.A. collected clinical information and L.N., I.M.S., L.M.N. and J.A.N. supervised the project. M.D. and J.A.N. conceived the project and H.J., M.F.L. and J.A.N. generated figures and wrote the paper. All authors read and approved the content of the manuscript.

## Additional information

**Competing interests:** The authors declare no competing financial interests

