## [Peer Review File · Nature Communications]

Reviewers' Comments:

Reviewer #1 (Remarks to the Author)

In their manuscript submitted to Nature Communications, entitled "IL-2 overrides the PD1/PD-L1 axis to enable complete tumor responses in an autologous immune-humanized melanoma mouse model" Jespersen et al suggest a simple hypothesis: that patients that will respond to therapy with adoptive cell transfer using tumor infiltrating lymphocytes (TIL) with IL-2 will gain no additional benefit from the use of a PD-1 abrogating antibody. They employ a novel system in which NOG mice, or NOG mice that are transgenic for IL-2 are implanted with tumor cells, and the post REP (rapid expansion) TIL are infused with IL-2 to try and generate a therapeutic effect. Some significant concerns arise about the use of the experimental system that make it difficult to make any real conclusions herein

For example, the data from figure 2 do not convincingly demonstrate that there is any association between the ability to generate a response to therapy with ACT in patients and the pattern of growth of TIL-treated tumors in the NOG mice. This is a severe defect of the experimental system, and completely limit the interpretation of any therapy experiments using TIL in the mouse model with or without the addition of any other agents, since the explanation of any failure to show benefit may be that the entire system is not a good surrogate for the outcome of treatment with TIL, not for example that PD-1 blockade failed to add to the benefit of ACT. The OS data are telling; there is little to suggest that TIL actually impact on OS at all in figure 2c. The authors appear to agree, since they state that "Taken together, TILs can delay tumor growth in some mice but cannot achieve regressions in NOG mice – even if the TILs came from patients that responded to adoptive cell transfer in the clinical trial." Then why use this specific system at all, since it appears to be a failed experimental model?

The use of 45000 U IL-2 daily in these experiments is well below that used in prior murine experiments in the 80s and 90s in mice, and well below what is used in humans, which represents the maximal tolerated dose given every 8 hours. This was recognized by the authors, who used an IL-2 transgenic mouse with high endogenous IL-2 to show that TIL persistence, known to be associated with positive outcome with that therapy, was improved with the higher levels of IL-2. This is a superior system to that used in figure 2.

The statement by the authors that "Third, anti-PD1 therapy does not further increase efficacy of ACT in the hIL2-NOG mouse model, arguing against the need of anti-PD1 therapy to enhance ACT in the clinic" does not make sense, because the authors have simply shown that if ACT with TIL in the presence of high doses of IL-2 induces a complete response, then the addition of pembrolizumab will not add anything. I would agree, and the conclusion is self evident and does not usefully add anything to our knowledge base.

Another statement by the authors that "A major finding of this paper is the apparent dependency of TILs on continuous supply of IL-2. This is notable considering the TIL therapy trials in patients that are currently performed to assess the appropriate dosing of IL-2" could have been confirmed by clinical data over a decade old showing that TIL have no human anti tumor activity without the addition of exogenous IL-2, found by the Surgery Branch quite a while ago.

Ultimately this work should describe the use of a new technology using a NOG/IL-2 transgenic mouse strain to grow tumors and successfully test ACT with human TIL as a test bed, with complete response in the mouse as a surrogate for regression in patients. That would require eliminating the frankly irrelevant data from figure 2, adding more patients to demonstrate a statistically important association between murine regression and human benefit since three patients do not make an important correlation, and the elimination of the verbiage about the use

of pembrolizumab since it make little sense to add anything to a complete response. The statement that PD-1 blockade does not add to the efficacy of TIL is incorrect, unjustified and not consistent with quite a few reports that patients failing one or even two checkpoint inhibitors have responded to ACT with TIL.

In detail:

The statement in the introduction "Cutaneous malignant melanoma can be cured by surgery but if the tumor cells metastasize the prognosis is dismal" is a bit dated and clinically naïve.

The statement 'Besides the obvious risk of contracting oncogenic mutations...' should be corrected; I doubt that such mutations are contagious.

Page 3 line 10 the spelling of ipilimumab should be corrected.

Reviewer #2 (Remarks to the Author)

The main claim of the paper is to develop and validate autologous immune-humanized melanoma mouse model available for preclinical efficacy assessment of immunotherapies. The second one is to highlight the fact that T-cell survival and antitumor activity require a continuous presence of IL-2. Various articles have already been reported in the development of humanized mouse models (for instance Sanmamed et al, Cancer Res 2015; Suarez et al, Oncotarget 2016). However, the novelty of the present article is to develop PDX models combining, in an autologous setting, tumor cells and tumor-infiltrating T cells (TILs) originating from the same patients. This approach presents the high interest to circumvent any allogeneic antitumor activity of T cells. Moreover, this article established a correlation between preclinical activity of "immunotherapy" and clinical activity observed in patients treated with anti-PD1 Mab (pembrolizumab). The paper might be of interested to any researchers working in the field of antitumor immunotherapies, as well as any clinician that are implicated or interested by recent clinical progress of immunotherapies largely but quiet recently published into cancer patients. Moreover, this paper develops a preclinical tool for specific assessments; it might therefore influence thinking in the field. Finally, the manuscript is well written.

However, few criticisms or questions have been raised and request detailed answers from the authors (see below).

1. Major comments and remarks:

- The model developed in the paper combines both human TILs and tumor cells injected into strongly immunodeficient mice. Such a situation might allow the occurrence of a graft versus host disease (GVHD) from human T cells against mouse cells. GVHD is mainly characterized in mice by weight loss, skin and hair abnormalities, sometime diarrhea, biological troubles and death. We can suppose that ex vivo expansion of TILs may reduce GVHD occurrence, but this remains unclear due to the fact that the authors never mentioned GVHD. Because of the impact of GVHD on the reproducibility and applicability of humanized mouse models, the authors are requested to respond to all above mentioned points.

- Immune reconstitution is well demonstrated by the results presented in the figure 3. Indeed, it was clearly shown, in the context of NOG-hIL2-TG mice, the intra-tumor accumulation of TILs labeled with a luciferase lentivirus. However, I regret the fact that no blood nor tumor assessment have been performed using flow cytometry analyses. Indeed, those studies might have defined the precise nature and phenotype of immune cells infiltrating the tumors, all informations of high interest in the development of an immunological mouse model. It could therefore be reasonable to ask the authors to perform new experiments in this sense, all experiments that could be add in a revised version of the manuscript.

- In the same point of view, it is not clear how immune reconstitution occurred in the data presented in the figure 2. This point constitutes a real difficulty to completely validate the author's interpretation.

2. Minor comments and remarks:

- There is an error in the figure 1 between d and e legends.
- The rationale of LDH determination as a surrogate marker of response is not clearly presented and, in fact, not really demonstrative (two of three resistant patients presented stable serum LDH levels).
- In the figure 2, it seems that responses are better in fast versus slow growing tumors. Can the authors comment this observation?
- Bioluminescence study are well-performed but quite difficult to evaluate. It is requested that the authors give precise quantification of luminescence for all experiments.
- Page 6, lines 16-18, the authors wrote: "..., and that PD-L1 mediated immune inhibition was abrogated with antibody treatment, ...": Can the authors explain this sentence and its rationale?
- The results presented in the figure 6D are over interpreted; a simple heterogeneity of tumor growth into mice is sufficient to explain various responses and should not be interpreted as sensitive or primary resistant tumor cells.

Rebuttal to Reviewers comments on Jespersen et al (NCOMMS-16-26192):

Reviewers comments in **black letters**, our responses in **blue**.

In their manuscript submitted to Nature Communications, entitled “IL-2 overrides the PD1/PD-L1 axis to enable complete tumor responses in an autologous immune-humanized melanoma mouse model“ Jespersen et al suggest a simple hypothesis: that patients that will respond to therapy with adoptive cell transfer using tumor infiltrating lymphocytes (TIL) with IL-2 will gain no additional benefit from the use of a PD-1 abrogating antibody. They employ a novel system in which NOG mice, or NOG mice that are transgenic for IL-2 are implanted with tumor cells, and the post REP (rapid expansion) TIL are infused with IL-2 to try and generate a therapeutic effect. Some significant concerns arise about the use of the experimental system that make it difficult to make any real conclusions herein

We thank the reviewer for taking the time and making effort of scrutinizing our study. We feel it has made the study clearer.

For example, the data from figure 2 do not convincingly demonstrate that there is any association between the ability to generate a response to therapy with ACT in patients and the pattern of growth of TIL-treated tumors in the NOG mice. This is a severe defect of the experimental system, and completely limit the interpretation of any therapy experiments using TIL in the mouse model with or without the addition of any other agents, since the explanation of any failure to show benefit may be that the entire system is not a good surrogate for the outcome of treatment with TIL, not for example that PD-1 blockade failed to add to the benefit of ACT. The OS data are telling; there is little to suggest that TIL actually impact on OS at all in figure 2c. The authors appear to agree, since they state that “Taken together, TILs can delay tumor growth in some mice but cannot achieve regressions in NOG mice – even if the TILs came from patients that responded to adoptive cell transfer in the clinical trial.” Then why use this specific system at all, since it appears to be a failed experimental model?

We thank the reviewer for helping us clarifying the message. We presented the data in Figure 2C to show that simply injecting TILs into NOG mice bearing autologous tumors will not cause rejection, even though the TILs do find the tumors (Figure 1). Given the firm, and correct, view of the reviewer that this is a failed experimental model we have decided to move the data in to Supporting Material. We do however want to disclose this data to PDX researchers to make them aware that the NOG model is not useful but we also agree with the reviewer that it takes up space. We want to reiterate that this model constituted our first albeit unsuccessful attempt at making a immune-humanized PDX model and that it served as the basis for developing the model that eventually turned out to better recapitulate the clinical situation upon ACT.

The use of 45000 U IL-2 daily in these experiments is well below that used in prior murine experiments in the 80s and 90s in mice, and well below what is used in humans, which represents the maximal tolerated dose given every 8 hours.

We were aware of the seminal mouse work by Rosenberg and colleagues showing that large doses of IL-2 are needed in mice. We also cite the study that IL-2 has a short half-life, which we reproduce in Figure 3A. Notably though, to avoid repeated injections of the fragile immunocompromised NOG mice we, before starting the project, reviewed the literature. We came up with the dose used for many valid reasons:

First, we found a paper by Rosenberg and colleagues (Kochenderfer et al. Blood 116:3875-3885, 2010) describing a dosing schedule that could support CAR-T cells. We followed this schedule in our study. Second, we performed calculations and compared to a published pharmacokinetics study in humans (Konrad et al Cancer Res 50:2009-2017, 1990):

- 1. Our dose 45000 U of aldesleukin (18×10^6 U/mg) gives 2h conc of approx 3 ng/ml = 54 U/ml serum (old Figure 2, new Figure S3).*
- 2. Bolus administration of 1×10^6 U/m²/6h gives mean concs of 40 U/ml (over 6h). After iv inj, peak conc are significantly higher (650 U/ml), but drop quickly (to ca 20U/ml after 2 h). IP or IM injections in humans have peak levels at 2h at ca 70 and 40 U/ml respectively). Hence, our measured dose is similar to that in humans.*

However, as noted in new Figure 2C, this dose was not sufficient to maintain viability and efficacy of TILs for the duration of the ACT experiment. When we found out that CIEA had developed an IL2-transgenic NOG mouse model we obtained this from Taconic while under early product development and repeated all our experiments, this time with a successful end result.

In the revised version we have made one more attempt at using injectable IL-2, out of interest and to honor the knowledgeable comment. To this end, we repeated the experiment with a different source of IL-2 (Peprotech) and injected daily for a prolonged time. Peprotech only provided an approximate unit (U) definition and when we use the same protocol as with the clinical grade we achieve higher plasma levels and we observe tumor regressions (New Supplemental Figure S5). On the other hand, as soon as we stop providing IL-2 the tumors re-grow. We therefore truly appreciate the added animal ethics value of the IL-2 transgenic mouse since we do not need to inject mice on a daily basis, which can be stressful for the animals if the tumors take a long time to become eradicated. An additional advantage is also that if this model is to be used to find out if non-responders can be converted into responders, additional treatments can be performed without needing to inject IL-2.

This was recognized by the authors, who used an IL-2 transgenic mouse with high endogenous IL-2 to show that TIL persistence, known to be associated with positive outcome with that therapy, was improved with the higher levels of IL-2. This is a superior system to that used in figure 2.

We thank the reviewer for appreciating this model.

The statement by the authors that “Third, anti-PD1 therapy does not further increase efficacy of ACT in the hIL2-NOG mouse model, arguing against the need of anti-PD1 therapy to enhance ACT in the clinic” does not make sense, because the authors have simply shown that if ACT with TIL in the presence of high doses of IL-2 induces a complete response, then the addition of pembrolizumab will not add anything. I would agree, and the conclusion is self evident and does not usefully add anything to our knowledge base.

Perhaps naïvely, we assumed that if the PD1/PD-L1 axis was a true barrier to the success of ACT we predicted that a) complete responses would occur faster when pembrolizumab was added together with TIL therapy, and b) some of the non-responders would become responders when pembrolizumab was added together with TIL therapy. We observed neither and when discussing the results with senior colleagues with long experience in immunoncology they informed us about earlier work demonstrating that IL-2 can override the negative impact of PD1/PD-L1 in vitro (Carter et al., Eur. J. Immunol. 2002. 32: 634–643). Whether or not this will hold true in the clinic would require randomized trials. Regardless, we have altered the title, abstract and discussion as to tone down this message and to not appear provocative. We have also clearly pointed out that the only thing we can say for sure with regards to anti-PD1 is that it does not convert a non-responder into a responder in the model and in the patient samples tested herein.

Another statement by the authors that “A major finding of this paper is the apparent dependency of TILs on continuous supply of IL-2. This is notable considering the TIL therapy trials in patients that are currently performed to assess the appropriate dosing of IL-2” could have been confirmed by clinical data over a decade old showing that TIL have no human anti tumor activity without the addition of exogenous IL-2, found by the Surgery Branch quite a while ago.

Yes, the clinical data are indeed there. We hope that our study is interpreted as very positive to ACT in general and the important studies made by the Surgery Branch in particular. Several of the co-authors of the current manuscript are themselves involved in the ACT clinical trials – the very same trials that have donated the samples to this study.

Ultimately this work should describe the use of a new technology using a NOG/IL-2 transgenic mouse strain to grow tumors and successfully test ACT with human TIL as a test bed, with complete response in the mouse as a surrogate for regression in patients. That would require eliminating the frankly irrelevant data from figure 2, adding more patients to demonstrate a statistically important association between murine regression and human benefit since three patients do not make an important correlation, and the elimination of the verbiage about the use of pembrolizumab since it make little sense to add anything to a complete response. The statement that PD-1 blockade does not add to the efficacy of TIL is incorrect, unjustified and not consistent with quite a few reports that patients failing one or even two checkpoint inhibitors have responded to ACT with TIL.

We have addressed most of these concerns above. We will move the data from Figure 2 to Supplemental information but wish to keep them for the mouse modeling community. We have toned down some of the statements about anti-PD1 but we actually regard our study to be consistent with the 'few reports that patients failing one or even two checkpoint inhibitors have responded to ACT with TIL'. In fact, since we, and others (Carter et al., Eur. J. Immunol. 2002. 32: 634–643), have argued that ACT and IL-2 may override PD1/PD-L1, hence our data is in accordance with that ACT could work in patients failing other immune therapies.

Regarding the amount of samples showing a correlation between murine regression and human benefit: We have put a significant effort into answering this point by recruiting new patient samples into the study and removed some of those that grew too slowly to be able to use in the IL2-transgenic NOG mice. We now have five different PDX models showing a perfect correlation between responses in patients and in IL2-trangenic NOG mice. The probability of this occurring by chance in five consecutive models is less than 5 % ($p=0.5^5=0.03125$). Our data therefore suggest that the PDXv2.0 mouse model is predictive for ACT in patients. We are very grateful for this reviewer comment since it has solidified the utility of the described mouse model.

In detail:

The statement in the introduction “Cutaneous malignant melanoma can be cured by surgery but if the tumor cells metastasize the prognosis is dismal“ is a bit dated and clinically naïve.

We apologize for this oversight. We agree that significant progress has been made and this should be highlighted in the introduction.

The statement ‘Besides the obvious risk of contracting oncogenic mutations...’ should be corrected; I doubt that such mutations are contagious.

We apologize for this error. We have changed ‘contracting’ to ‘causing’.

Page 3 line 10 the spelling of ipilimumab should be corrected.

We apologize for this error and have corrected it.

Reviewer #2:Expert in Patient derived xenografts
(Remarks to the Author):

The main claim of the paper is to develop and validate autologous immune-humanized melanoma mouse model available for preclinical efficacy assessment of immunotherapies. The second one is to highlight the fact that T-cell survival and antitumor activity require a continuous presence of IL-2. Various articles have already been reported in the development of humanized mouse models (for

instance Sanmamed et al, Cancer Res 2015; Suarez et al, Oncotarget 2016). However, the novelty of the present article is to develop PDX models combining, in an autologous setting, tumor cells and tumor-infiltrating T cells (TILs) originating from the same patients. This approach presents the high interest to circumvent any allogeneic antitumor activity of T cells. Moreover, this article established a correlation between preclinical activity of “immunotherapy” and clinical activity observed in patients treated with anti-PD1 Mab (pembrolizumab). The paper might be of interested to any researchers working in the field of antitumor immunotherapies, as well as any clinician that are implicated or interested by recent clinical progress of immunotherapies largely but quiet recently published into cancer patients. Moreover, this paper develops a preclinical tool for specific assessments; it might therefore influence thinking in the field. Finally, the manuscript is well written.

We thank the reviewer for the kind words about or work.

However, few criticisms or questions have been raised and request detailed answers from the authors (see below).

1. Major comments and remarks:

- The model developed in the paper combines both human TILs and tumor cells injected into strongly immunodeficient mice. Such a situation might allow the occurrence of a graft versus host disease (GVHD) from human T cells against mouse cells. GVHD is mainly characterized in mice by weight loss, skin and hair abnormalities, sometime diarrhea, biological troubles and death. We can suppose that ex vivo expansion of TILs may reduce GVHD occurrence, but this remains unclear due to the fact that the authors never mentioned GVHD. Because of the impact of GVHD on the reproducibility and applicability of humanized mouse models, the authors are requested to respond to all above mentioned points.

We have worked with PDX models for several years and have seen GVHD in some mice transplanted with biopsies from human tumors. We are therefore well adversed with the phenotypes of GVHD. At variance with other models (e.g. Sanmamed et al, Cancer Res 2015) we did not observe GVHD in any of the mice in this study. We have added a section in the discussion where we speculate that this is due to at least three reasons: i) the tumor cells transplanted does not contain immune cells, ii) the TILs have been selected in the patients to recognize the tumor cells and may therefore be specific, and iii) the TIL cultures do not contain B-cells, macrophages or other immune cells that also constitute major culprits in GVHD (Lockridge et al., Biol Blood Marrow Transplant. 2013, 19:1310-22; Covassin et al., Clin Exp Immunol. 2013, 174:372-88). We hope our model will become a foundation on which to build new models and test immune therapies that can convert a non-responder to e.g. ACT into a responder.

- Immune reconstitution is well demonstrated by the results presented in the figure 3. Indeed, it was clearly shown, in the context of NOG-hIL2-TG mice, the intra-tumor accumulation of TILs labeled with a luciferase lentivirus. However, I regret the fact that no blood nor tumor assessment have been performed using flow cytometry analyses. Indeed, those studies might have defined the precise

nature and phenotype of immune cells infiltrating the tumors, all informations of high interest in the development of an immunological mouse model. It could therefore be reasonable to ask the authors to perform new experiments in this sense, all experiments that could be add in a revised version of the manuscript.

We have performed an immune phenotyping analysis of all TILs prior to infusion (new Figure S1) and of a tumor and blood of a responder (new Figure S4). We thank the reviewer for this suggestion and it was a reasonable request that yielded interesting results for those that wish to reproduce our work in the lab and find biomarkers for response in the clinic. The analyses we made in response to this comment have contributed to making this a stronger manuscript.

- In the same point of view, it is not clear how immune reconstitution occurred in the data presented in the figure 2. This point constitutes a real difficulty to completely validate the author's interpretation.

We have performed immunohistochemistry to address this. Please note that the previous Figure 2 has been moved to become Supplemental Figure S3 to answer concerns from reviewer 1. We have added in the new immunohistochemistry images to demonstrate that immune reconstitution is very poor in ordinary NOG mice.

2. Minor comments and remarks:

- There is an error in the figure 1 between d and e legends.

We apologize for this error and have corrected it.

- The rational of LDH determination as a surrogate marker of response is not clearly presented and, in fact, not really demonstrative (two of three resistant patients presented stable serum LDH levels).

We agree completely and in the revised version we have presented spider plots and swimmer plots in Figure S2A. These demonstrate more comprehensively how the patients responded to TIL therapy.

- In the figure 2, it seems that responses are better in fast versus slow growing tumors. Can the authors comment this observation?

We cannot explain this more than if the control grows fast, separation of the curves will be faster if there is a response. Overall, response in the clinic is a better predictor of effect in mice than growth speed.

- Bioluminescence study are well-performed but quiet difficult to evaluate. It is requested that the authors give precise quantification of luminescence for all experiments.

We have enhanced the visibility of the numbers in the scale bars where it is relevant. We agree that the fonts were too small to evaluate.

- **Page 6, lines 16-18, the authors wrote: “..., and that PD-L1 mediated immune inhibition was abrogated with antibody treatment, ...”: Can the authors explain this sentence and its rationale?**

Thank you for allowing us to rephrase the sentence; we agree it was not well written.

- **The results presented in the figure 6D are over interpreted; a simple heterogeneity of tumor growth into mice is sufficient to explain various responses and should not be interpreted as sensitive or primary resistant tumor cells.**

We apologize for not making this clear enough. The resistant cell is indeed resistant. We had omitted the in vitro data showing that this clone, which appeared when single-cell cloning of the melanoma culture was performed after viral transduction, is resistant to TILs. In response to reviewer 1 we have recruited more samples of patients treated with TILs. Given the surplus of the new data we think that the resistant clone does not add anything to the study and we therefore removed these data from the figures and the text.

Reviewers' comments:

Reviewer #1 (Remarks to the Author):

In their manuscript re-submitted to Nature Communications, entitled "IL-2 overrides the PD1/PD-L1 axis to enable complete tumor responses in an autologous immune-humanized melanoma mouse model" Jespersen et al suggest that they have developed a new system that comprises a test bed for the evaluation of TILs in a murine model that employs a novel system in which NOG mice that are transgenic for IL-2 are implanted with tumor cells, and the post REP (rapid expansion) TIL are infused with IL-2 to try and generate a therapeutic effect.

The authors previously had used a NOG mouse with adoptive transfer of TIL and exogenous IL-2 to try and show that there was an association between the ability to generate a response to therapy with ACT in patients and the pattern of growth of TIL-treated tumors in the NOG mice. That indeed did not work, and they have relegated these negative data to the appendix since it is a precursor to the important work that they now emphasize.

One concern that was expressed was that in their original system the use of 45000 U IL-2 daily in these experiments was well below that used in prior murine experiments in the 80s and 90s in mice, and well below what is used in humans, which represents the maximal tolerated dose given every 8 hours. The authors response to this concern was not really relevant, and they have never done experiments with TID higher doses of IL-2 to demonstrate that higher doses of IL-2 might make the regular NOG model work. They do point out that there is tumor regression in the regular NOG model with long term IL-2, but never show us that this altered NOG model is not suitable for prediction of outcome with TIL.

The idea by the authors that anti-PD1 therapy does not further increase efficacy of ACT in the hIL2-NOG mouse model did not make sense previously, and it still does not; the authors could have taken one of their suboptimal models with low dose IL-2 as in 4b in which there was only a modest effect of TIL on tumor regression and added PD-1 antibody to see if it added anything to survival; instead they again chose to add it to a curative maneuver with TIL so that no possible improvement could be seen.

It was previously felt that this work should have been re-configured as a manuscript that described the use of a new technology using a NOG/IL-2 transgenic mouse strain to grow tumors and successfully test ACT with human TIL as a test bed, with complete response in the mouse as a surrogate for regression in patients. That would require a significantly larger number of tumors implanted in NOG-IL-2 mice and treated with autologous TIL to get a reliable estimate of the positive and negative predictive power of the assay, and an assessment of persistence of the TIL as well. How many patients to test would depend on a statistical review, but would need to be greater than simply 3 responders and 2 non-responders that resulted in a borderline p value.

Reviewer #2 (Remarks to the Author):

Reviewer #2:Expert in Patient derived xenografts (Remarks to the Author):

The revised version of the article has largely been improved and might therefore be planned for publication. However, very few comments persist, as detailed in the following point-by-point analysis :

- The model developed in the paper combines both human TILs and tumor cells injected into strongly immunodeficient mice. Such a situation might allow the occurrence of a graft versus host disease (GVHD) from human T cells against mouse cells. GVHD is mainly characterized in mice by weight loss, skin and hair

abnormalities, sometime diarrhea, biological troubles and death. We can suppose that ex vivo expansion of TILs may reduce GVHD occurrence, but this remains unclear due to the fact that the authors never mentioned GVHD. Because of the impact of GVHD on the reproducibility and applicability of humanized mouse

models, the authors are requested to respond to all above mentioned points.

Authors' response: We have worked with PDX models for several years and have seen GVHD in some mice transplanted with biopsies from human tumors. We are therefore well advised with the phenotypes of GVHD. At variance with other models (e.g. Sanmamed et al, Cancer Res 2015) we did not observe GVHD in any of the mice in this study. We have added a section in the discussion where we speculate that this is due to at least three reasons: i) the tumor cells transplanted does not contain immune cells, ii) the TILs have been selected in the patients to recognize the tumor cells and may therefore be specific, and iii) the TIL cultures do not contain B-cells, macrophages or other immune cells that also constitute major culprits in GVHD (Lockridge et al., Biol Blood Marrow Transplant. 2013, 19:1310-22; Covassin et al., Clin Exp Immunol. 2013, 174:372-88). We hope our model will become a foundation on which to build new models and test immune therapies that can convert a non-responder to e.g. ACT into a responder.

New comment: the aim of my remark was not to teach phenotype of GVHD but to invite the authors to indicate in the article few symptoms that have been or not observed. The strict minimum would be indication of the mice weight during treatments, that constitute a requested follow-up of any in vivo experiments. Those data should be added in a supplementary figure. Finally, I invite the authors to mention more cautiously their conclusions in the Discussion section regarding the absence of GVHD.

- Immune reconstitution is well demonstrated by the results presented in the figure 3. Indeed, it was clearly shown, in the context of NOG-hIL2-TG mice, the intra-tumor accumulation of TILs labeled with a luciferase lentivirus. However, I regret the fact that no blood nor tumor assessment have been performed using flow cytometry analyses. Indeed, those studies might have defined the precise nature and phenotype of immune cells infiltrating the tumors, all informations of high interest in the development of an immunological mouse model. It could therefore be reasonable to ask the authors to perform new experiments in this sense, all experiments that could be add in a revised version of the manuscript.

Authors' response: We have performed an immune phenotyping analysis of all TILs prior to infusion (new Figure S1) and of a tumor and blood of a responder (new Figure S4). We thank the reviewer for this suggestion and it was a reasonable request that yielded interesting results for those that wish to reproduce our work in the lab and find biomarkers for response in the clinic. The analyses we made in response to this comment have contributed to making this a stronger manuscript.

New comment: OK.

- In the same point of view, it is not clear how immune reconstitution occurred in the data presented in the figure 2. This point constitutes a real difficulty to completely validate the author's interpretation.

Authors' response: We have performed immunohistochemistry to address this. Please note that the previous Figure 2 has been moved to become Supplemental Figure S3 to answer concerns from reviewer 1. We have added in the new immunohistochemistry images to demonstrate that immune reconstitution is very poor in ordinary NOG mice.

New comment: OK.

- There is an error in the figure 1 between d and e legends.

We apologize for this error and have corrected it. OK

- The rationale of LDH determination as a surrogate marker of response is not clearly presented and, in fact, not really demonstrative (two of three resistant patients presented stable serum LDH

levels).

Authors' response: We agree completely and in the revised version we have presented spider plots and swimmer plots in Figure S2A. These demonstrate more comprehensively how the patients responded to TIL therapy.

New comment: OK, but the issue concerning the rationale remains undefined.

- In the figure 2, it seems that responses are better in fast versus slow growing tumors. Can the authors comment this observation?

We cannot explain this more than if the control grows fast, separation of the curves will be faster if there is a response. Overall, response in the clinic is a better predictor of effect in mice than growth speed. OK

- Bioluminescence study are well-performed but quite difficult to evaluate. It is requested that the authors give precise quantification of luminescence for all experiments.

We have enhanced the visibility of the numbers in the scale bars where it is relevant. We agree that the fonts were too small to evaluate. OK

- Page 6, lines 16-18, the authors wrote: "..., and that PD-L1 mediated immune inhibition was abrogated with antibody treatment, ...": Can the authors explain this sentence and its rationale? Thank you for allowing us to rephrase the sentence; we agree it was not well written. OK

- The results presented in the figure 6D are over interpreted; a simple heterogeneity of tumor growth into mice is sufficient to explain various responses and should not be interpreted as sensitive or primary resistant tumor cells.

We apologize for not making this clear enough. The resistant cell is indeed resistant. We had omitted the in vitro data showing that this clone, which appeared when single-cell cloning of the melanoma culture was performed after viral transduction, is resistant to TILs. In response to reviewer 1 we have recruited more samples of patients treated with TILs. Given the surplus of the new data we think that the resistant clone does not add anything to the study and we therefore removed these data from the figures and the text. OK

Reviewers' comments:

Reviewer #1 (Remarks to the Author):

In their manuscript re-submitted to Nature Communications, entitled “IL-2 overrides the PD1/PD-L1 axis to enable complete tumor responses in an autologous immune-humanized melanoma mouse model” Jespersen et al suggest that they have developed a new system that comprises a test bed for the evaluation of TILs in a murine model that employs a novel system in which NOG mice that are transgenic for IL-2 are implanted with tumor cells, and the post REP (rapid expansion) TIL are infused with IL-2 to try and generate a therapeutic effect.

The authors previously had used a NOG mouse with adoptive transfer of TIL and exogenous IL-2 to try and show that there was an association between the ability to generate a response to therapy with ACT in patients and the pattern of growth of TIL-treated tumors in the NOG mice. That indeed did not work, and they have relegated these negative data to the appendix since it is a precursor to the important work that they now emphasize.

Yes, we thank the reviewers for the comments that helped us to correctly emphasize our study.

One concern that was expressed was that in their original system the use of 45000 U IL-2 daily in these experiments was well below that used in prior murine experiments in to 80s and 90s in mice, and well below what is used in humans, which represents the maximal tolerated dose given every 8 hours. The authors response to this concern was not really relevant, and they have never done experiments with TID higher doses of IL-2 to demonstrate that higher doses of IL-2 might make the regular NOG model work. They do point out that there is tumor regression in the regular NOG model with long term IL-2, but never show us that this altered NOG model is not suitable for prediction of outcome with TIL.

We are in complete agreement with the reviewer. If more IL-2 is injected during a prolonged time we can achieve regression of tumors also in NOG mice. This is an advantage for those investigators that would wish to use ACT in mice but have not yet or cannot breed the *hIL2*-NOG mice. Obviously we could have come to this conclusion earlier had we only realized that the IL-2 dosing we found in the literature was the problem with the first attempts (original figure 2 and now sup fig 3). However, we think that the *hIL2*-NOG mice as a model is much more convenient than injections of IL-2 since it saves research efforts and time, mouse handling and money. Therefore, the utility of the *hIL2*-NOG mice is the main message of the manuscript and we do not feel it is meaningful for us to perform more experiments using injectable IL-2.

The idea by the authors that anti-PD1 therapy does not further increase efficacy of ACT in the *hIL2*-NOG mouse model did not make sense previously, and it still does not; the

authors could have taken one of their suboptimal models with low dose IL-2 as in 4b in which there was only a modest effect of TIL on tumor regression and added PD-1 antibody to see if it added anything to survival; instead they again chose to add it to a curative maneuver with TIL so that no possible improvement could be seen.

Again, we are in agreement with the reviewer. We have even further toned down the message. We do not state that anti-PD1 would not benefit patients treated with ACT as we cannot prove this. The reviewer suggests that we should have 'taken one of their suboptimal models with low dose IL-2' but we respectfully would like to point out that we provided data that this will not work. In Supplemental Figure 5A-B we report the experiments that led us to realize that there is a threshold of IL-2 needed for the therapeutic response. Incidentally two of the low IL-2 mice were treated also with pembrolizumab and they did not respond. We think the better approach is what we have done in this revised version of the text: to mention in the discussion that the PD1 data could represent a lack of a component in the model.

It was previously felt that this work should have been re-configured as a manuscript that described the use of a new technology using a NOG/IL-2 transgenic mouse strain to grow tumors and successfully test ACT with human TIL as a test bed, with complete response in the mouse as a surrogate for regression in patients.

Yes, this is the main message of the paper.

That would require a significantly larger number of tumors implanted in NOG-IL-2 mice and treated with autologous TIL to get a reliable estimate of the positive and negative predictive power of the assay, and an assessment of persistence of the TIL as well. How many patients to test would depend on a statistical review, but would need to be greater than simply 3 responders and 2 non-responders that resulted in a borderline p value.

We have added one more model that was very slow growing in early passages in mice. This model came from a patient that did not respond and we have now shown that neither did the mice. At this time we have now used the models with matched tumors and TILs that we could get to grow in a reasonable time.

To assess if the model is predictive, the calculations provided in the revised version discussed with two biostatisticians. One of them (Joakim Larsson) performed permutations on the probability that 3/6 models would responding at that these are the very same models that came from the 3/6 patients that responded to ACT. The p-value in favor of that the samples matched because they were linked, and not by chance, was 0.015. The other biostatistician (Erik Larsson) used a hyper-geometric test (Fisher's) and obtained a p value of 0.05 of co-occurrence.

During the discussion a concern was raised over the fact that we had performed experiments on samples from patients we knew before if they respond or not to ACT. In this way, the patients' responses were predictive of responses in mice, not the opposite.

Therefore it is better to say that the responses correlate in patients and in the mouse model. We have changed this in the text. A larger study, using a prospective approach, will be initiated. This study may be published in a few years time and will be able to clearly establish the predictive power of the PDXv2.0 model.

Reviewer #2 (Remarks to the Author):

Reviewer #2:Expert in Patient derived xenografts (Remarks to the Author):

The revised version of the article has largely been improved and might therefore be planned for publication.

We thank the reviewer for the previous comments and for appreciating our efforts to respond to them in the revised version.

However, very few comments persist, as detailed in the following point-by-point analysis:
New comment: the aim of my remark was not to teach phenotype of GVHD but to invite the authors to indicate in the article few symptoms that have been or not observed. The strict minimum would be indication of the mice weight during treatments, that constitute a requested follow-up of any in vivo experiments. Those data should be added in a supplementary figure. Finally, I invite the authors to mention more cautiously their conclusions in the Discussion section regarding the absence of GVHD.

We thank the reviewer for this comment, as it is very important. Since we did not observe any reproducible signs of GVHD we assumed it not to be a major problem. However, we have just found an accepted manuscript not yet listed on PubMed. This study aims to develop a mouse model of GVDH by transplanting human T cells into NOG mice. It turns out that CD4+ T cells are real culprits in this model, causing skin phenotypes, alopecia, weight loss and death. However, in *hIL2*-NOG mice, CD8+ cells also result in a lethal phenotype by 8 weeks after injection but the authors do not show that this is due to GVDH as opposed to a lympho-proliferative disease owing to expansion of T-cells (Fig 4 of their manuscript, see below.). As seen in the Figure 2 of our manuscript, T cells that do not have a target tumor will proliferate and spread to spleen and bone marrow. However, if there is a tumor they will predominantly home to the tumor and attempt to eradicate it. We believe that T cell education and homing is the reason behind the lack of a detrimental effect of T cells in our mouse model. Please note that we show experiments that lasted for much longer than 8 weeks (Fig 3. MM33 and Fig 4 MM11 and a new model MM04) so we think we can model ACT in a time frame that allows viability of the mice and escape the potential detrimental effects of T cell expansion or GVDH. But of course, we agree completely that it needs to be further elaborated on in the Discussion and we thank for the kind invitation to do so. We have also added the epub manuscript in the reference list to guide those wishing to use PDXv2.0 for studies primarily on the role of CD4+ in ACT.

Fig. 4

Figure 4 from Ito et al (epub ahead of print). A-B) weight and survival of mice transplanted with CD8+ cells. C-D) amount and phenotype of of CD8+ cells in NOG or hIL2-NOG mice.

Rest of our responses comments were approved by reviewer 2.

REVIEWERS' COMMENTS:

Reviewer #1 (Remarks to the Author):

The authors have responded adequately to the comments of the reviewers, although the response about the use of PD-1 blockade in a sub optimal model of ACT is still not appropriate. Instead of a suboptimal level of IL-2, a lower number of TIL could be used with pembro to test the idea that it adds to the efficacy of ACT. Nonetheless, this should simply be added as a comment to the work.

Reviewer #2 (Remarks to the Author):

Just one issue persisted after the first review. I thank the authors that have completely responded to this last issue with own weight data and new published article mention.

I therefore consider that the article is now appropriate for publication.

Reviewers' comments:

Reviewer #1 (Remarks to the Author):

The authors have responded adequately to the comments of the reviewers, although the response about the use of PD-1 blockade in a sub optimal model of ACT is still not appropriate. Instead of a suboptimal level of IL-2, a lower number of TIL could be used with pembro to test the idea that it adds to the efficacy of ACT. Nonetheless, this should simply be added as a comment to the work.

We thank the reviewer all the constructive comments that helped us to correctly emphasize our study. We have now added a comment in Discussion that a lower amount of TILs could be used to assess the effects of checkpoint inhibitors such as anti-PD1.

Reviewer #2 (Remarks to the Author):

Just one issue persisted after the first review. I thank the authors that have completely responded to this last issue with own weight data and new published article mention.

I therefore consider that the article is now appropriate for publication.

We thank the reviewers for the comments and for all the kind words.